# The Influence of Combined Pruning and the Use of Root Application of Two Biostimulants and Foliar Nutrition on the Growth and Flowering of Panicle Hydrangea Plants

Sławomir Świerczyński [1],[*] and Ilona Świerczyńska [2]

1   Department of Ornamental Plants, Dendrology and Pomology, Poznan University of Life Sciences, J.H. Dąbrowskiego 159, 60-594 Poznań, Poland
2   Institute of Plant Protection—National Research Institute, Władysława Węgorka 20, 60-318 Poznań, Poland; i.swierczynska@iorpib.poznan.pl
*   Correspondence: slawomir.swierczynski@up.poznan.pl; Tel.: +48-61848-7955

**Abstract:** The aim of this experiment was to assess how the interaction of two treatments influenced the growth and flowering of two varieties of Panicle hydrangea shrubs. The first treatment was plant pruning. Simultaneously, the plants received one of the three following treatments: root application of *Trichoderma atroviride*, root application of BlackJak biostimulant, or foliar application of a multi-component fertilizer. Simultaneous pruning and inoculation of the plants with the *Trichoderma atroviride* mycelium improved the length of hydrangea shoots the most, as compared with the control plants (18%). These two treatments also increased the number of flowers (16–47%, depending on the variety) and the fresh weight of plants (10–28%) compared with the control plants. *T. atroviride* alone improved the number of flowers in both varieties (19–24%) and the diameter of inflorescences in the 'Silver Dollar' one (17%). The foliar nutrition increased the fresh weight of plants by 7–57%, depending on the cultivar and pruning. It also increased the diameter and number of inflorescences in one of the varieties. Together with pruning, it intensified the growth of shoots in both cultivars (5–10%). The BlackJak biostimulant treatment gave ambiguous results. In combination with pruning, it improved the length of shoots (15%) in one cultivar and the fresh weight in the other (18%). Without pruning, the treatment increased the number of flowers (16%) and the diameter of inflorescences (9%) in one cultivar. It increased the fresh weight of plants in both cultivars (19–21%). Regardless of the other treatments, pruning increased the length of the shoots and the fresh weight of the plants. On the other hand, it reduced the number of flowers and their diameter. In most cases, the biostimulant treatment and foliar fertilization improved the growth and flowering of the plants. In combination with pruning, they improved the growth of the hydrangea shrubs but reduced the number and diameter of flowers. The simultaneous Ta treatment and pruning were the most beneficial for the growth and flowering of the panicled hydrangea plants grown in containers in a nursery.

**Keywords:** *Hydrangea paniculata*; varieties; container cultivation; nursery; additional care treatments

## 1. Introduction

Hydrangeas are some of the most popular ornamental plants in gardens because of their showy blooms with different sizes and shapes of flowers [1]. They are easy to care for and cause few maintenance problems with pests and diseases. They tolerate shade and adapt well to both acidic and alkaline soils [2]. The Hydrangea genus comprises approximately 80 species [3]. In recent years, Panicle Hydrangea (*Hydrangea paniculata* Siebold) has become one of the most popular species due to its characteristic white inflorescences, long flowering period, a wide range of growing locations, relatively good frost resistance, and low susceptibility to diseases [4]. The attractive appearance and numerous varieties of Panicle Hydrangea encourage growers to increase its production in nurseries [5]. However, when these plants are grown in a nursery, shrubs have few inflorescences and shoots, which

are significantly elongated and do not look attractive. Another difficulty is the fact that customers are not interested in purchasing them in the second half of summer because they do not produce flowers. To prevent these shortcomings, young shrubs can undergo early-stage pruning during production, leading to a subsequent round of flowering, as evidenced by a previous experiment. Pruning is a simple procedure, widely used in the production of fruit and vegetables. It can promote the formation of branches and stimulate vegetative growth, thus increasing the yield [6]. However, this simple procedure is not popular in the cultivation of ornamental plants because it is not known how it will affect specific species. Current scientific studies indicate that pruning may disturb the balance between vegetative and generative growth of plants [7]. When shoots are pruned, more dormant buds are activated, resulting in a greater leaf area and dry weight of plants [8–10]. Below the cutting site, the GA3 content in the buds increases, whereas the IAA and ABA content decreases, which promotes the formation and development of buds [11]. The evidence confirming these opinions comes from the experiment on the Malus 'Profusion' species [12]. It showed that unpruned trees bloomed the worst, whereas heavily pruned trees produced more vigorous and flowering shoots. In another experiment, *Loropetalum chinense* 'Rubrum' was pruned to varying degrees [13]. Pruning made the plants bloom faster and produce more flowers. Moderate pruning of jasmine forced the plants to flower and increased the number of flowers [14]. When rose shoots were cut at the base to a length of 10 cm, their quality increased, and the flowering time of the plants shortened [15]. Also, the pruning of hybrid roses increased the number of flowers on the plants by 68.7% [16].

In order to improve the growth of ornamental plants, various species of fungi of the Trichoderma genus are used, which are widely recognized as plant growth stimulants [17–20]. These fungi have been widely studied as biological control agents [21]. Trichoderma-based preparations are used to protect plants against pathogens [22]. They are known to improve plant resistance to stress, such as drought, by increasing the branching capacity of the root system, thereby improving nutrient uptake and water acquisition [23–26]. Trichoderma fungi are most commonly applied to soil and leaves. The latter method is more effective, especially in preventing soil-borne diseases [27]. However, these authors did not find any improvement in the growth of young olive trees grown in containers after the foliar application of the Trianum P biostimulant based on *T. harzianum* fungi. They observed increased growth of young trees only after using the slow-release granular fertilizer Osmocote. Doni et al. [28], Bhandari et al. [29], and Andrzejak and Janowska [30] used Trichoderma and observed improvement in the height of different ornamental plants and rice, an increased photosynthesis rate and stomatal conductance and higher chlorophyll content in the leaves.

Hydrangeas are generally considered to have high nutrient requirements, especially nitrogen, to support their vigorous growth [31,32]. Standard soil application of fertilizers in nursery production can also be supported with appropriate biostimulants. Annual hydrangea plants exposed to water stress were treated with seaweed-based biostimulants in an attempt to reduce the stress [33]. The plants of *Hydrangea paniculata* treated with the biostimulants had the same number of branches and the same fresh weight as those that were not exposed to the stress. Thanks to the foliar application of the Asahi biostimulant to hydrangea 'Anabelle' shrubs, it was possible to reduce the dose of the Osmocote fertilizer [34].

The BlackJak (Bioagris, Poland) biostimulant containing humic acids has so far been applied to the ornamental plant *Euphorbia* x *lomi* [35] as well as to cherry trees [36] and grape shrubs [37]. The growth and quality of the plants treated in these experiments varied. The aim of our experiment was to determine whether it was justified to introduce. Experiments with this biostimulant showed that it improved the weight of lettuce plants and the chlorophyll content in leaves [38]. Jasim [39] observed that when broccoli leaves were sprayed with a biostimulant containing humic acid, the diameter and weight of broccoli flower heads, as well as the number and area of leaves, increased.

The aim of our experiment was to determine whether it was justified to introduce two additional care treatments to nursery production and to examine their mutual impact

on the growth and flowering of *Hydrangea paniculata* plants. The first treatment was pruning. The other one was the root application of *Trichoderma atroviride* mycelium (Ta) and the BlackJak (BJ) biostimulant, as well as the foliar application of Universol Green (UG) fertilizer.

## 2. Materials and Methods

### 2.1. Plant Material and Growth Conditions

The experiment was conducted in a private nursery between 2022 and 2023 as part of research at the Faculty of Agriculture, Horticulture and Bioengineering, University of Life Science in Poznań. Two-year-old Panicle Hydrangea shrubs were planted in containers in March 2021 and 2022. The study included two varieties: 'Silver Dollar' and SKYFALL 'Frenne', sourced from a nursery in the Netherlands. The factorial experiment was arranged in a random block design, with two factors, namely pruning (shoot pruning or no pruning) and four types of plant treatments, resulting in a total of eight experimental combinations. The plant treatments included (i) *Trichoderma atroviride* mycelium; (ii) BlackJak biostimulant; (iii) Universol® Green foliar fertilizer, with a balanced NPK formula with magnesium and trace elements, applied as a foliar fertilizer; and (iv) no treatment, serving as the control. Each experimental combination was replicated 20 times for each variety. Two biostimulants were applied to the soil in mid-April 2023. The foliar fertilizer was applied immediately after pruning. The treatment was repeated after 3 weeks. The plants were hand-sprayed, with portable protective curtains placed between the combinations. The doses, form of application, and content of active ingredients of the tested biostimulants and fertilizer are listed in Table 1. In each of the four treatments, 20 plants were pruned, and 20 were not pruned. The cut was made in mid-June at the height between the first and second internode under the inflorescence. The plants were arranged in parallel rows to minimize differences in the effects of lighting and irrigation with angle sprinklers (Figure 1). They were grown in a nursery in 5-L containers. Deacidified peat from Agaris, fraction 0/40 mm, was used for planting. The pH of the substrate ranged from 5.5 to 6.0. The substrate was enriched with Osmocote® Exact Standard [16 N-9 P-12 K + 2 Mg and trace elements] 3–4 mm fertilizer at a dose of 3 g·L$^{-1}$ of substrate. The plants were constantly watered with sprinklers with a water dose of 8 L·m$^{-2}$ on days without rainfall. Koppert's Spical preparation, containing the California red spider mite (*Neoseiulus californicus*), was applied to protect the plants against red spider mites (*Tetranychus urticae*). Weeds in containers and near plants were regularly removed manually.

**Table 1.** Plant treatments used in the experiment.

| Treatment | Concentration Dose per Plant | Application Form | Composition |
|---|---|---|---|
| Trichoderma atroviride | 10 mL | root application | spore-forming mycelium of the genus *Trichoderma atroviride* |
| BlackJak® Bioagris, Poland | 0.5 mL·L$^{-1}$ 300 mL per plant | root application | leonardite: min. 28%; organic substances: min. 20%; humins; ulmic acids; humic acids; fulvic acids |
| Universol® Green | 2 g·L$^{-1}$ 40 mL per plant | foliar application | N 23%, K 8.3%, P 2.6%, Fe 0.1%, Cu 0.1%, Zn 0.1%, Mn 0.4%, B 0.01%, Mo 0.01% |

Plant measurements were taken at the end of October due to persistent flowers and the plant's suspended growth. The total shoot length (TSL, cm) and the diameter of inflorescences (DI, cm) were measured for all plants, the number of inflorescences (NI) per plant was counted, and the fresh weight of the plants (FWP, g) after shaking them from the substrate was weighed. The results of hydrangea growth and flowering presented in the tables are the averages from two years of experiments (2022–2023).

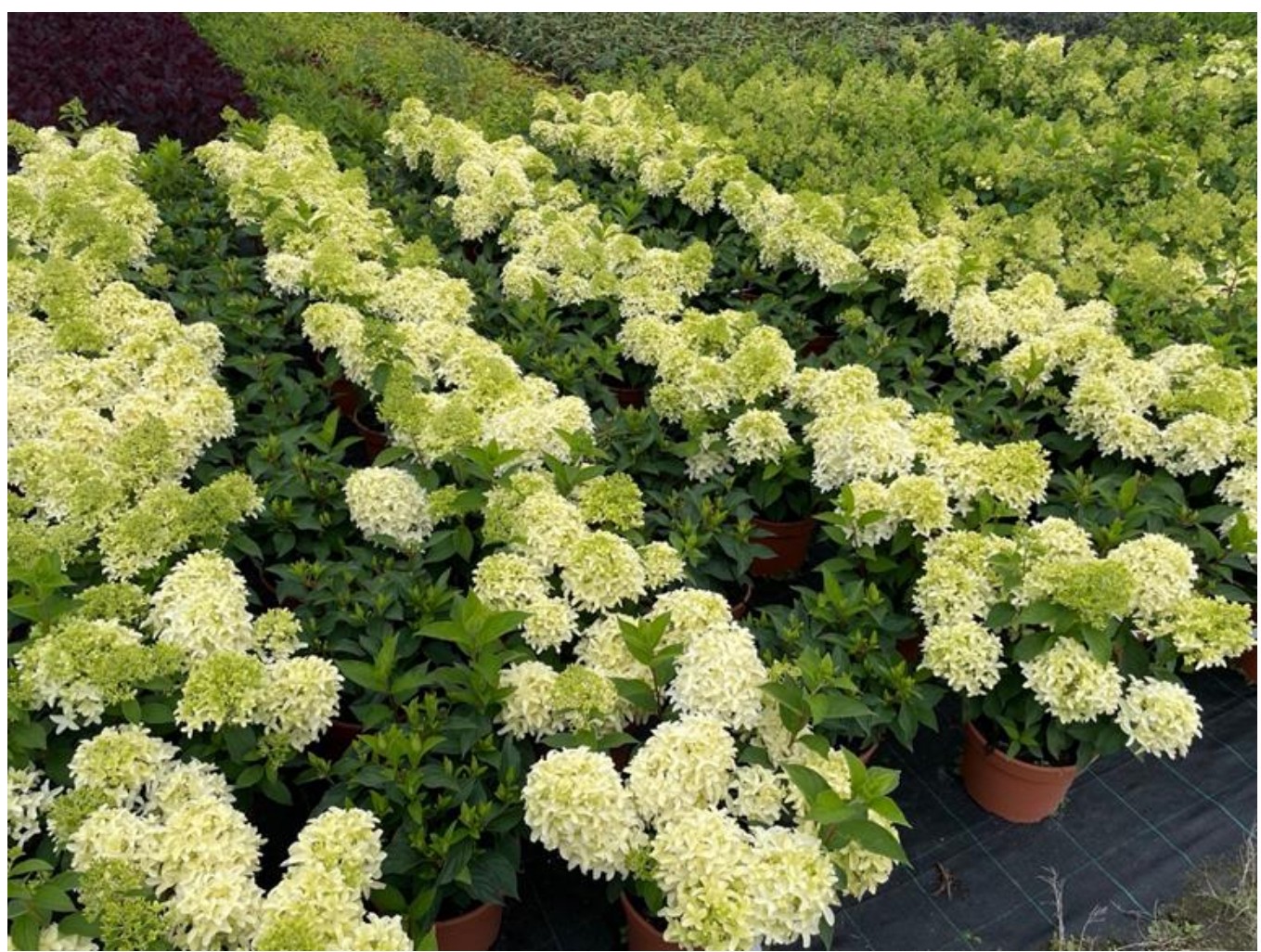

**Figure 1.** Plants pruned and unpruned (with flowers) of Panicle Hydrangea in experiment.

*2.2. Laboratory Analysis of the Presence of the Fungus*

In order to confirm the presence of Trichoderma fungi in the rhizosphere of the examined hydrangea plants, root samples were taken for mycological analysis. Healthy roots were surface disinfected for half a minute in 5% sodium hypochlorite and then rinsed with distilled water. After drying, the roots were cut with a sterile lancet into pieces several millimeters long. The prepared plant material was placed on a PDA [Potato Dextrose Agar] culture medium in Petri dishes [90 mm]. The incubation process took place at 21 °C. After 6 days of cultivation, the presence of Trichoderma spp. colonies, as well as colonies of other filamentous fungi and yeasts, was confirmed (Figure 2). Fungi of the Trichoderma species were identified based on the assessment of macroscopic and microscopic features using mycological keys.

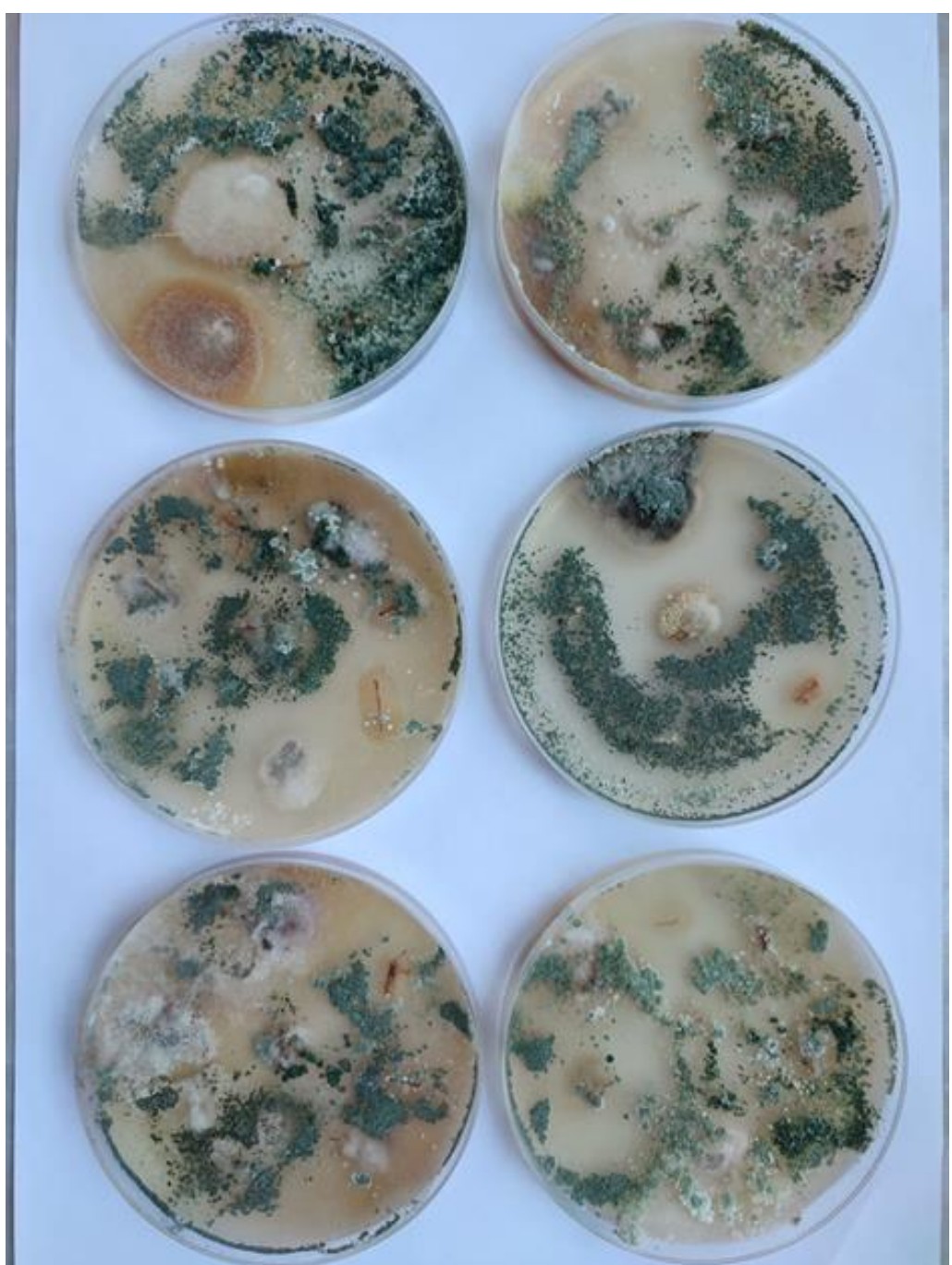

**Figure 2.** Colonies of the fungus Trichoderma originating from root samples of hydrangea plants treated with this fungus (phot. Świerczyńska I.).

*2.3. Data Analysis*

The results of the study were analyzed using STATISTICA 13.1 [Statsoft Polska, Kraków, Poland]. To perform statistical calculations of the obtained results, a two-factor analysis of variance [treatment and pruning] was used for each variety separately. The Duncan test was used, with a significance level of $\alpha = 0.05$.

## 3. Results

*3.1. Analysis of Plant Growth and Flowering*

Pruning significantly intensified the growth of the plants (Tables 2 and 3). After the treatment of pruned plants of both cultivars with Ta mycelium, the TSL reached the highest

level. The other two treatments (BJ and UG) also gave better results than the control. Only the biostimulant based on humic substances did not improve the growth of SKYFALL 'Frenne' plants. The biostimulants and foliar fertilizer did not cause differences in the parameters of the unpruned plants.

**Table 2.** Interaction of pruning and the use of biostimulants or foliar fertilizer on the total length of shoots of 'Silver Dollar' Panicle Hydrangea (cm).

| Treatments | No Pruning | Shoot Pruning | Average |
|---|---|---|---|
| Trichoderma atroviride | 765.0 [a] | 1103.7 [e] | 934.3 [c] |
| BlackJak | 739.7 [a] | 1036.3 [d] | 888.0 [b] |
| Universol Green | 762.3 [a] | 950.0 [c] | 856.2 [b] |
| Control | 741.3 [a] | 901.0 [b] | 821.2 [a] |
| Average | 752.1 [a] | 997.8 [b] | |

Mean values marked with the same letters do not differ significantly at the level of significance $\alpha = 0.05$, using Duncan's test.

**Table 3.** Interaction of pruning and the use of biostimulants or foliar fertilizer on the total length of shoots of SKYFALL 'Frenne' Panicle Hydrangea (cm).

| Treatments | No Pruning | Shoot Pruning | Average |
|---|---|---|---|
| Trichoderma atroviride | 340.0 [a] | 1325.0 [d] | 832.5 [c] |
| BlackJak | 367.3 [a] | 1103.7 [b] | 735.5 [ab] |
| Universol Green | 347.0 [a] | 1188.0 [c] | 767.5 [b] |
| Control | 304.3 [a] | 1084.3 [b] | 694.3 [a] |
| Average | 339.7 [a] | 1175.3 [b] | |

Mean values marked with the same letters do not differ significantly at the level of significance $\alpha = 0.05$, using Duncan's test.

Both with and without pruning, the highest NI of 'Silver Dollar' plants was achieved after the inoculation with Ta. The other two treatments did not change NI significantly. All treatments applied to the unpruned plants of the second cultivar SKYFALL 'Frenne' gave better results than in the control variant. After the inoculation with mycelium and spraying with the UG fertilizer, the pruned plants had a better NI. Pruning significantly reduced the NI of both cultivars (Tables 4 and 5).

**Table 4.** Interaction of pruning and the use of biostimulants or foliar fertilizer on the number of inflorescences of 'Silver Dollar' Panicle Hydrangea.

| Treatments | No Pruning | Shoot Pruning | Average |
|---|---|---|---|
| Trichoderma atroviride | 17.2 [e] | 9.0 [b] | 13.1 [b] |
| BlackJak | 13.5 [c] | 4.9 [a] | 9.2 [a] |
| Universol Green | 15.4 [d] | 5.2 [a] | 10.3 [a] |
| Control | 13.9 [cd] | 4.8 [a] | 9.4 [a] |
| Average | 13.0 [b] | 6.0 [a] | |

Mean values marked with the same letters do not differ significantly at the level of significance $\alpha = 0.05$, using Duncan's test.

**Table 5.** Interaction of pruning and the use of biostimulants or foliar fertilizer on the number of inflorescences SKYFALL 'Frenne' of Panicle Hydrangea.

| Treatments | No Pruning | Shoot Pruning | Average |
|---|---|---|---|
| Trichoderma atroviride | 12.0 [e] | 6.8 [b] | 9.4 [c] |
| BlackJak | 11.7 [d] | 5.9 [a] | 8.5 [b] |
| Universol Green | 10.9 [d] | 6.8 [b] | 8.8 [b] |
| Control | 10.1 [c] | 5.7 [a] | 7.9 [a] |
| Average | 11.0 [b] | 6.3 [a] | |

Mean values marked with the same letters do not differ significantly at the level of significance α= 0.05, using Duncan's test.

The pruned plants of the 'Silver Dollar' variety had a significantly larger DI after inoculation Ta. The other two treatments also improved the DI in the pruned plants as compared with the control. After the inoculation and treatment with the foliar fertilizer, the results of the unpruned plants were better than in the control variant. The pruned plants of the second variety were characterized by a larger DI only after foliar fertilization. Regardless of the treatment, the unpruned plants of both cultivars had a larger DI than the pruned ones (Tables 6 and 7).

**Table 6.** Interaction of pruning and the use of biostimulants or foliar fertilizer on the diameter of inflorescences of 'Silver Dollar' Panicle Hydrangea (cm).

| Treatments | No Pruning | Shoot Pruning | Average |
|---|---|---|---|
| Trichoderma atroviride | 11.9 [e] | 6.8 [b] | 9.4 [c] |
| BlackJak | 11.1 [d] | 5.9 [a] | 8.5 [b] |
| Universol Green | 10.9 [d] | 6.8 [b] | 8.8 [b] |
| Control | 10.2 [c] | 5.7 [a] | 7.9 [a] |
| Average | 11.0 [b] | 6.3 [a] | |

Mean values marked with the same letters do not differ significantly at the level of significance α= 0.05, using Duncan's test.

**Table 7.** Interaction of pruning and the use of biostimulants or foliar fertilizer on the diameter of inflorescences of SKYFALL 'Frenne' Panicle Hydrangea (cm).

| Treatment | No Pruning | Shoot Pruning | Average |
|---|---|---|---|
| Trichoderma atroviride | 11.7 [c] | 7.6 [a] | 9.6 [a] |
| BlackJak | 11.9 [c] | 8.1 [a] | 10.0 [a] |
| Universol Green | 13.1 [d] | 9.0 [b] | 11.0 [b] |
| Control | 12.4 [cd] | 7.5 [a] | 10.0 [a] |
| Average | 12.3 [b] | 8.0 [a] | |

Mean values marked with the same letters do not differ significantly at the level of significance α= 0.05, using Duncan's test.

The fertilizer treatment resulted in the highest FW of the unpruned plants of the 'Silver Dollar' variety. In comparison with the control variant, the other two treatments also increased the FW of the plants. The four treatments applied to the pruned plants did not result in significant differences between them. The pruned plants of the second variety inoculated with mycelium had the highest weight. However, in the absence of pruning, the highest plant weight was recorded when the foliar fertilizer was applied. Also, the other two treatments increased the weight of both pruned and unpruned plants of the second variety which was greater than in the control variant. The FW of the pruned plants of both

varieties was significantly greater than that of the unpruned ones, regardless of the other treatments (Tables 8 and 9).

**Table 8.** Interaction of pruning and the use of biostimulants or foliar fertilizer on the weight of plants of 'Silver Dollar' Panicle Hydrangea (g).

| Treatments | No Pruning | Shoot Pruning | Average |
|---|---|---|---|
| Trichoderma atroviride | 396.0 [b] | 563.0 [d] | 479.5 [bc] |
| BlackJak | 351.0 [b] | 553.7 [d] | 452.3 [b] |
| Universol Green | 455.3 [c] | 548.7 [d] | 502.0 [c] |
| Control | 290.0 [a] | 509.0 [d] | 399.5 [a] |
| Average | 373.1 [a] | 543.6 [b] | |

Mean values marked with the same letters do not differ significantly at the level of significance α= 0.05, using Duncan's test.

**Table 9.** Interaction of pruning and the use of biostimulants or foliar fertilizer on the weight of plants of SKYFALL 'Frenne' Panicle Hydrangea (g).

| Treatments | No Pruning | Shoot Pruning | Average |
|---|---|---|---|
| Trichoderma atroviride | 376.3 [b] | 550.0 [f] | 463.2 [c] |
| BlackJak | 365.0 [b] | 468.7 [de] | 416.8 [b] |
| Universol Green | 431.0 [cd] | 492.3 [e] | 461.7 [c] |
| Control | 307.3 [a] | 398.0 [bc] | 352.7 [a] |
| Average | 369.9 [a] | 477.3 [b] | |

Mean values marked with the same letters do not differ significantly at the level of significance α= 0.05, using Duncan's test.

### 3.2. Analysis of the Presence of the Trichoderma Fungus

*Trichoderma atroviride* colonies originating from the root samples of the plants treated with this fungus were characterized by a rapid growth rate and abundant sporulation. The color of the fungus colony was usually a shade of green—from pale green to intense green, occasionally with a gray or yellow coating (Figure 2). The colonies of the fungus in question showed that the inoculation of the plants was effective.

## 4. Discussion

### 4.1. Plant Growth and Flowering after Applying the Pruning

Pruning had the greatest influence on the growth of the plants. All treatments of the pruned plants of the 'Silver Dollar' variety resulted in a significantly greater TSL than in the unpruned plants. For the SKYFALL 'Frenne' variety, the value of this parameter in the pruned plants inoculated with the mycelium was almost three times greater than in the unpruned ones. Pruning also improved the FW of the plants. The experiment showed that pruning stimulated the growth of the plants and increased their vegetative mass. On the other hand, it reduced the number of flowers in both varieties, where the 'Silver Dollar' had more flowers than the SKYFALL 'Frenne' variety. The diameter of the inflorescences was smaller in pruned plants and larger in the case of the SKYFALL 'Frenne' variety. In comparison with the results of other experiments [5], the DI of the unpruned plants of *Hydrangea paniculata* was similar. Other researchers observed similar positive effects of pruning on the growth of ornamental plants. Zhang et al. [13] applied different types of pruning of *Loropetalum chinense* 'Rubrum' plants and then compared the relationship between the pruning intensity and the number of flowers of plants. The research showed that pruning increased the number of flowers. Hassanein [16] found that the number of flowers in pruned rose bushes was 68.7% greater than in the control plants. However, in our experiment, pruning reduced the number of flowers because flower buds in this plant

develop on the previous year's shoots. Pruning reduces the number of flowers that bloom later. Additionally, the diameter of the inflorescences may have been reduced due to the loss of some of the assimilates accumulated in the cut-off shoots.

### 4.2. Growth and Flowering of Plants after Application of Trichoderma Atroviride

Our experiment showed that the inoculation of plants with *Trichoderma atroviride* improved their growth and flowering. The pruned plants of both varieties treated with the mycelium had the highest TSL. Yahya et al. [40] observed a similar relationship in *Lantana camara* plants. The plants treated with *T. harzianum* and pruned most severely grew taller and had greater trunk diameters than those pruned less heavily and not inoculated. On the contrary, Cruz et al. [41] did not observe any effect of *Trichoderma* spp. on the length of shoots or the flower quality of gladiolus. In our experiment, the inoculated plants had the most flowers, regardless of pruning. The inoculated plants in both the pruned and unpruned groups were characterized by the greatest fresh mass, but this effect was observed only in one of the cultivars. The combination of two treatments, i.e., pruning and inoculation, gave the best results.

The effect of the Trichoderma spp. fungi has been tested on various species, e.g., Begonia × tuberhybrida [30] and *Kalanchoe* sp. [42]. Experiments proved the positive effect of this treatment on the number of flowers. Additionally, Sisodia et al. [43] observed that the treatment had extended the flowering period of selected Gladiolus varieties. Moreover, during the propagation of the GiSelA 6 rootstock, the *Trichoderma harzianum* fungi improved the growth of the root system and shoots [44]. When the fungi were applied to annual cucumber plants [45] and lettuce [46], they stimulated more intensive growth of their shoots and roots. According to Lorito et al. [24], the mechanisms responsible for the beneficial effects of Trichoderma spp. and stimulation of plant growth have not been fully elucidated and are based on the suggestion that the stimulation results from increased availability of nutrients. Some researchers claim that after applying Trichoderma sp., the growth of plants depends on their species or even the cultivar. For example, Fiorentino et al. [47] found that two strains of Trichoderma spp. significantly improved the growth parameters of lettuce, but not rocket. Tucci et al. [48] found that the inoculation of cultivated and wild tomato varieties with *T. atroviride* and *T. harzianum* improved their yield but in a variety-dependent manner. The results of experiments also depended on the species of Trichoderma sp. fungi used, as was observed in interrelations *Trichoderma* spp. on soybean [49]. Additionally, Di Marco and Osti [50], concluded that the beneficial effects of *Trichoderma* spp. may be related to the choice of the nursery stage production and the way of application. Their study showed that a reduced dose did not reduce the activity of *Trichoderma* spp. Overall, all types of Trichoderma applications stimulated the growth of the grape root system. Also, an experiment with *Trichoderma atroviride* applied to the roots of young poplar trees grown in containers showed that the treatment increased the height, trunk diameter, as well as dry weight of the roots and shoots [51]. Similar effects were observed in our experiment after root application of *T. atroviride* to the Panicle Hydrangea roots. This method of application is much more effective than the foliar treatment used by Di Vaio et al. [27], who did not observe increased growth of young olive trees cultivated in containers in a nursery. On the other hand, Rakibuzzaman et al. [52] observed an increase in the dry weight of tomatoes after inoculation. They attributed it to the combined effect of increased leaf area index, nutrient uptake, and photosynthetic intensity.

### 4.3. Growth and Flowering of Plants after Foliar Spraying with Fertilizer

In our experiment, foliar fertilization alone had a positive effect on the fresh weight of the plants and the diameter of their inflorescences. Additionally, in combination with pruning, it increased the growth of shoots. In another study [53], foliar fertilization significantly improved the growth of maiden apple trees in a nursery. On the other hand, in another experiment, half the dose of a soil fertilizer was combined with foliar nutrition [54], but the content of macroelements in the leaves of maiden apple trees did not change significantly.

However, the foliar application of biostimulants and fertilizers to maiden sweet cherry trees positively affected their gas exchange parameters [55]. The effects observed after the foliar fertilization of hydrangeas lead to the conclusion that even when an optimal dose of fertilizer is applied to soil, foliar treatment is recommended in nursery production, preferably in combination with pruning. Pruning reduces the content of some of the nutrients accumulated in the shoots. On the other hand, foliar fertilization partially compensates for this loss.

*4.4. Growth and Flowering of Plants after Root Application of BlackJak Preparation*

The leonardite-based preparation used for the treatment of plants in our experiment had the least effect on the growth and flowering of plants compared to the two above-mentioned treatments. However, it significantly improved some growth parameters, which were better than in the control variant. Humic acids are known to affect the growth and development of plants [56–58]. Demiren [36] observed that they improved the nutrition of cherry fruits. Hajizadeh et al. [59] found that they improved the yield of sugar beets by 7%, while Olego et al. [34] observed no improvement in the size and yield of grapes. In our experiment, BlackJak biostimulant was applied with pruning ambiguously to improve the shoot length and fresh mass of the plants, one cultivar only. When the biostimulant was applied without pruning the number and diameter of inflorescences increased only in the plants of one cultivar again. However, the fresh weight of the plants of both varieties increased. Further research on the application of biostimulants containing humic acids into soil in nursery practice is necessary due to the ambiguity of our research findings (positive results for one cultivar only).

**5. Conclusions**

The most beneficial effect was observed when pruning was combined with the root application of *Trichoderma atroviride*. The two treatments improved the growth of the two hydrangea varieties grown in container cultivation in a nursery. The foliar fertilization most effectively influenced the growth of plants, especially their fresh mass and, to a lesser extent, their flowering. The biostimulant BlackJak caused smaller changes in flowering and plant growth than the other two treatments, but it improved most of the parameters under analysis. Pruning increased the growth of shoots and the fresh mass of the plants, but it had a negative effect on the number of flowers and their diameter. The plants of the 'Silver Dollar' cultivar had more flowers, whereas the SKYFALL 'Frenne' plants had larger diameters of inflorescences.

**Author Contributions:** S.Ś. and I.Ś. contributed to the study's conception and design. Material preparation, data collection, and analysis were performed by S.Ś. and I.Ś. The first draft of the manuscript was written by S.Ś., and all authors commented on the previous version of the manuscript. All authors have read and agreed to the published version of the manuscript.

**Funding:** Publication was co-financed within the framework of the Polish Ministry of Science and Higher Education.

**Data Availability Statement:** Data are contained within the article.

**Conflicts of Interest:** The authors declare no conflicts of interest.

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
