# Peer review of "The Influence of Combined Pruning and the Use of Root Application of Two Biostimulants and Foliar Nutrition on the Growth and Flowering of Panicle Hydrangea Plants"

_agronomy, doi:10.3390/agronomy14040687_

Round 1

Reviewer 1 Report

Comments and Suggestions for Authors

The manuscript presents an interesting study on the combined effect of pruning, Trichoderma, a bio-stimulant, and foliar fertilizer on Hydrangea plants. However, I have some comments for the authors:

Lines 114 and 115: Details about the type of sprayer used for foliar application and how contamination between treatments was avoided would be helpful.

Line 123: More details about the contents of Osmocote are needed.

Lines 135 and 136: Are the average data from the years 2022 and 2023? Please clarify for better understanding.

Line 180: It is suggested to modify the font size in Table 4 for better readability.

Line 230: The observations about the roots mentioned are not presented in the results, which may cause confusion.

Lines 245 and 246: It would be useful to include a hypothesis explaining why pruning led to a reduction in the number of flowers.

General comment: The authors focus much of the discussion on comparing their results with those of other authors, however, it would be beneficial to add a hypothesis explaining the results obtained in their research.

Author Response

Thank you very much for all your kind comments, which were taken into account in the submitted publication after corrections.

Reviewer 2 Report

Comments and Suggestions for Authors

Overall, the work is relatively simple from an experimental standpoint, but it is presented in a confusing and non-linear manner. The discussion illogically references a series of papers on vastly different species, ranging from flowering plants like kalanchoe to cherry rootstocks, then to cucumber and lettuce, tomato, soybean, grape, poplar, apple, sweet cherry, and sugar beet. The discussion is consistently disorganized in discussing the effects of various treatments on flower numbers, root and shoot growth, yield, plant height, dry weight, and fruit size; propagation compared to cultivation. It also haphazardly suggests possible effects related to species, variety, nutrient uptake, and photosynthetic activity.

Several works are cited, many of which are not relevant to the current experiment, but a critical discussion with an explanation and evaluation of results, along with pertinent literature supporting the general conclusions, is lacking. The conclusions are also contradictory.

The authors need to reformulate the discussion by comparing the various results with experimental research that can either support or contradict these findings.

Moreover, authors must refer to a professional manuscript editing service or to native English speaker for improving the English in the manuscript.

Below are some specific notes:

- Line 34: Panicle Hydrangea is already in the title; the common name can be replaced here with the Latin name Hydrangea paniculata.

- Lines 50-51: It is unclear if after the first flowering, and if this has already been experimented.

- Lines 77-82: It's odd that the authors chose a study on olive when there are several studies, e.g., on chrysanthemum or ornamental species (see for example Bhandari et al 2021 - An overview of multifaceted role of Trichoderma spp. for sustainable agriculture; Roman Andrzejak and Beata Janowska 2022 - Trichoderma spp. Improves Flowering, Quality, and Nutritional Status of Ornamental Plants), which have issues more related to hydrangea than fruit tree species.

- Line 87: What biostimulants?

- Lines 92: BlackJak is a brand, so company references and patents must be given.

- Lines 92-98: Here, fruit species are mixed with leafy vegetables and broccoli, effect on production with chlorophyll content of leaves, etc. This part is poorly written.

- Line 101: Why is "Hydrangea" capitalized when the common name Panicle Hydrangea is mentioned?

- Lines 102-104: The authors mention a second treatment that actually consists of three different treatments. It would be more accurate to discuss the effect of pruning (pruning treatment) and the effect of plant biostimulants and foliar fertilizer (treatment with Trichoderma and leonardite as biostimulants and a water-soluble versatile fertilizer).

- The Materials and Methods section needs to be more precise, indicating more clearly when the planting was done, which factors were considered, how many repetitions were considered, how many plants per repetition, and when treatments were applied.

- The two commercial products should also be briefly described in the text to improve readability, for example: "A versatile fertilizer, Universol® Green, was used, having a nitrogen-based N:K ratio (2:1) or … having balanced NPK formula with magnesium and trace elements."

- Line 132: The term 'sum of the length of shoots' is awkward. Replace with 'total shoot length (TSL) measured as …...'. Abbreviations can be used: ID = inflorescence diameter, IN = inflorescence number per plant, TFW = total fresh weight (roots and above-ground biomass).

- Lines 135-136: This sentence is unclear, and the results should not be presented in the Materials and Methods section.

- Line 159: TSL

- Line 161: It's not ideal to repeatedly mention the commercial name of the product used. It could be referred to as a biostimulant based on humic substances or a biostimulant with high concentration of humic substances, or a product based on humic extracts with acidic pH, etc.

In general, it would be desirable to use more appropriate language and avoid continuous repetitions of the same term.

- Lines 163-164: Incorrect sentence.

- Tables: The data reported are indeed the interaction of the 'pruning' factor x 'biostimulant/fertilizer treatment' factor, so the table title needs to be modified accordingly.

For example: Interaction effect of pruning (P) x applied biostimulant/fertilizer (B/F) on total shoot lenght (cm) of ‘Silver Dollar’ panicle hydrangea.

Additionally, indicate whether significance should be interpreted within columns.

- Lines 173-179: The writing is grammatically incorrect. The entire text needs careful revision.

- Lines 229-230: What types of observations on nursery roots?

- Lines 234-235: Which experiments and on which species? The reader should not have to refer to the bibliography to understand what the author means.

- Lines 235-236: What research? What is stated is well-known, so more commentary on this aspect is needed.

- Lines 239-242: This part is confusing. These results are contrary to those obtained in the present experiment. The authors may write, for example: "Contrary to what was highlighted by our experiment, Zhang et al. found that...". Then explain the possible reasons for these discrepancies.

- Lines 240-242: Incorrect sentence.

- Lines 243-246: A repetition that adds nothing.

- Line 253: In our experiment...

- Lines 256-263: Unclear.

- Line 265: Independent???

- Line 268: What is meant by "longer flowering"?

- The discussion section should be articulated separately, addressing aspects such as the effect as a biofertilizer, nutrient availability and related biological processes, effect on soil properties and microbial activities, effect of species and cultivar, etc.

- Line 327: Flowering is not improved by plant pruning.

These are more of a repetition of the results than conclusions.

Comments on the Quality of English Language

Authors must refer to a professional manuscript editing service or to native English speaker for improving the English in the manuscript.

Author Response

Thank you very much for all your kind comments, which were taken into account in the submitted publication after corrections. Language correction by a specialist is in progress. However, in accordance with the publisher's suggestion, I am sending the publication after making the changes proposed by the reviewers regarding the methodology.

When it comes to articulating the separately studied impact factors, it was done by dividing the discussion into the impact of pruning, biostimulants and foliar fertilization. The influence of the variety was not examined separately, each of them was considered statistically independently. The influence of soil properties, biological processes, etc. was not examined, therefore they were not addressed in the discussion. The research will be extended to include these parameters according to the reviewer's suggestions and presented in a future publication.

Round 2

Reviewer 2 Report

Comments and Suggestions for Authors

The authors have implemented the changes taking into account the suggestions. However, some sections still appear unclear, and therefore, I provide further advice below to enhance the comprehension and fluency of the text.

Regarding the tables, I recommend modifying them as indicated in the attached PDF file.

Lastly, concerning the text itself, I once again advise engaging professional English language reviewers to improve the writing.

L 51-53: To prevent these shortcomings, young shrubs can undergo early-stage pruning during production, leading to a subsequent round of flowering as evidenced by a previous experiment.

L 86: Which species of leaves? It would be helpful to specify whether they are ornamental species, flowering species, or other categories.

L 117-119: Two-year-old Panicle Hydrangea shrubs were planted in containers in March 2021 and 2022. The study included two varieties: 'Silver Dollar' and SKYFALL 'Frenne', sourced from a nursery in the Netherlands.

L 120-126: Still not clearly described. I would suggest modifying as follows:

The factorial experiment was arranged in a ………. design, with two factors, namely pruning (shoot pruning or no pruning) and four types of plant treatment, resulting in a total of eight experimental combinations. The plant treatments included: i) Trichoderma atroviride mycelium; ii) BlackJak biostimulator; iii) Universol® Green foliar fertilizer, having balanced NPK formula with magnesium and trace elements, applied as a foliar fertilizer; and iv) no treatment, serving as the control. Each experimental combination was replicated 20 times for each variety.

Line 143: Table 1 . Plant treatments used in the experiment.

(Moreover, for completeness, I would add in table 1 the manufacturer and any applicable copyright symbol ©)

L 183: for tables, see attached pdf

Comments on the Quality of English Language

The English form must still be improved

Author Response

All very important comments of the reviewer have been taken into account, thank you very much. A significant change in the quality of the English language was also carried out by a qualified person